# Red Cell Distribution Width as a Predictive Factor of Celiac Disease in Middle and Late Adulthood and Its Potential Utility as Celiac Disease Screening Criterion

**DOI:** 10.3390/ijerph20010066

**Published:** 2022-12-21

**Authors:** Julia María Cabo del Riego, María Jesús Núñez-Iglesias, José Paz Carreira, Andrés Blanco Hortas, Tamara Álvarez Fernández, Silvia Novío Mallón, Sofía Zaera, Manuel Freire-Garabal Núñez

**Affiliations:** 1Clinical Analysis Laboratory, Department of Immunology, Lucus Augusti University Hospital, 27003 Lugo, Spain; 2Doctoral Programme in Medicine Clinical Research, International PhD School of the University of Santiago de Compostela (EDIUS), 15782 Santiago de Compostela, Spain; 3SNLLaboratory, School of Medicine and Dentistry, University of Santiago de Compostela, 15782 A Coruña, Spain; 4Department of Psiquiatry, Radiology, Public Health, Nursing and Medicine, University of Santiago de Compostela, 15782 A Coruña, Spain; 5Department of Hematology, Oncology Center of Galicia, 15009 A Coruña, Spain; 6Health Research Institute Foundation (FIDIS) of Santiago de Compostela, Lucus Augusti University Hospital, 15706 Santiago de Compostela, Spain; 7Department of Pharmacology, Pharmacy and Pharmaceutical Technology, University of Santiago de Compostela, 15782 Santiago de Compostela, Spain

**Keywords:** red blood width, celiac disease, onset, diagnosis, adult, elderly

## Abstract

Red cell distribution width (RDW) could be of interest by its potential use in the assessment of celiac disorder (CD). The main objective of this study was to evaluate the case positive rate of CD and the utility of red cell distribution width (RDW) in the CD diagnosis. This prospective study included 9.066 middle adult (≥45 years old) and elderly patients (≥60 years old) from 2012 to 2021. CD diagnosis was performed by CD antibody tests (serology and Human Leucocyte Antigen genotype (HLA)) and biopsy. Gastrointestinal and extra-intestinal manifestations as well as hematological and biochemical parameters were analyzed. CD diagnoses were confirmed in 101 patients (median (IQR) age = 62 (52.3–73); 68.32% women) by serologic tests (100%) and intestinal biopsy (88.12%), showing mainly marked or complete atrophy (76.24%, MARSH 3a–c). Anemia was the most commonly presenting extra-intestinal manifestation (28.57%). Among 8975 individuals without CD, 168 age and sex matched were included. By comparison of CD and no CD individuals, we observed that high >14.3% RDW was exhibited by 58.40% and 35.2% individuals with CD and without CD, respectively. Furthermore, high RDW is associated with CD and grade III atrophy. We suggest that RDW could be used as a CD screening criterion.

## 1. Introduction

Celiac Disease (CD) is known to be an autoimmune disease that affects children and young adults. In recent years, its appearance in middle- and late-adulthood has gained interest. It is important to note that a quarter and a fifth of CD is diagnosed at or after the age 60 and 65, respectively [1]. 

Mention should be made of the fact that the non-specificity of the manifestations in late adulthood may be responsible for not being diagnosed in 60% of patients [1]. Recognizing symptoms and clinical signs of CD, including extra-intestinal ones such as anemia, is an important step. 

Notably, anemia is recorded in up to 50% of newly diagnosed CD. Furthermore, anemia may be the only manifestation of CD [1]. Against this background, it has to be stated that misrecognition of the association of extra-intestinal manifestations such as anemia and CD could delay its diagnosis years [2]. The multifactorial etiology of the high prevalence of anemia in middle and late adulthood can provide a major contribution to under-diagnosed CD [3]. Additionally, there are differences in testing for CD based on age or gender. Thus, young men are the most frequently tested [4]. 

Of major concern is the increasing incidence of CD among adults and elderly and the CD-related morbidity [1,5]. To this situation is added the fact that anemia is considered a geriatric syndrome [6]; being responsible for morbidity and mortality [6,7]. Likewise, minimally invasive serum screening tests for CD should be recommended in these patients. As part of these screening tests, red blood cell distribution width (RDW) became of particular interest as it is a routine and available test that could be useful in predicting CD [8,9,10], irrespective of anemia status [8]. Our objectives were to evaluate the case positive rate of CD in middle and late adulthood as well as to identify RDW as a predictive factor of CD and its relationship with diagnosis criteria.

## 2. Materials and Methods

### 2.1. Study Design and Subjects

Patients were recruited at the Lucus Augusti University Hospital (Galicia, Spain), from January of 2012 to December of 2021. 9066 patients ≥45 years old attended at Galician Healthcare Service (SERGAS) in whom there was a suspicion of CD were referred from primary care physician, geriatrician or gastroenterologist to Laboratory of Immunology at the Lucus Augusti University Hospital in order to confirm CD diagnoses. 

### 2.2. Clinical Evaluation

The CD onset was categorized as gastrointestinal manifestations, extra-intestinal manifestations, and associated disorders (diabetes, autoimmune thyroidal disease, systemic autoimmune disease, etc.) as well as family history of CD [11].

### 2.3. Celiac Disease Antibody Tests: Serology and Human Leucocyte Antigen Genotype (HLA)

All serological tests performed are accredited by the UNE-EN-ISO 15,189 standards for clinical laboratories (accreditation May 2011; reaccreditation 2013, 2015, 2017, 2019, 2020, and 2021). 

Serology and HLA genotype were performed according to the Ministry of Health protocol [12]. At first, we determined serum IgA and Tissue transglutaminase antibody (IgA) (TTG2-IgA). Total serum IgA was analyzed using BNII nephelometry (Siemens BNII, Oststeinbek, Germany). The indicative reference values 70 mg/dL were supplied by the manufacturer’s recommendations and guidelines. The EliA Celikey Kit (ThermoFisher Scientific, Waltham, MA, USA) was used to determine serum TTG2-IgA by means of an enzyme-linked immunosorbent assay. Cut-off was >8 U/mL and <2 U/mL for positivity and negativity, respectively. Between 2 and 8 (U/mL), considered to be a grey zone, endomysial antibodies IgA (EMA-IgA) were determined and a follow-up test was established.

In the case of patients who exhibited selective IgA deficiency, the deamidated gliadin peptide antibody (IgG) (DGP-IgG) was determined as the first step (GGP Kit, Thermo Fisher Scientific, Waltham, MA, USA). Additionally, at least one additional IgG class test (TTG2-IgG and EMA-IgG) was performed. Both tests were done by automated fluoroenzyme INMUNOCAP 250 instrument (Thermo Fisher Scientific) according to the manufacturer’s instructions. Positive and negative control samples were analyzed in each run. Our laboratory is involved in inter-laboratory comparison (laboratory quality standards) to ensure the comparability and acceptability of testing results among the laboratories that integrate this program (United Kingdom National External Quality Assessment Scheme (UK NEQAS) and Quality Club (ThermoFisher, Friburgo, Germany)). 

For TTG2-IgA values >2 U/mL EMA-IgA were measured. Serum EMA- IgA titers were measured by indirect immunofluorescence (AESKU slides, AESKU-Diagnostics, Wendelsheim, Germany). EMA assays were read by 2 experienced observers. A dilution 1:10 was considered positive and positive sera was further diluted from 1:10 to 1:2560. 

HLA genotyping was performed using the single specific primer polymerase chain reaction (SSPPCR) DQ kits DQA1*05, DQB1*02, DQA1*0301, DQB1*0302, DQA1*0505, DQB1*0202 for detecting the DQ2.5, DQ2.2, and DQ8 haplotypes (Celiacstrip HLA DQ2DQ8 OPERON, Immune and Molecular Diagnostics, Zaragoza, Spain).

### 2.4. Biochemical and Hematological Laboratory Parameters

All biochemical tests performed are accredited by the UNE-EN-ISO 15,189 standards for clinical laboratories (accreditation 2011; reaccreditation 2013, 2015, 2017, 2019, 2020, and 2021). All tests were performed in accordance with the manufacturer’s recommendations. Calibrators were provided as part of the assay (Siemens^®^) and an external Quality Control (QC) was performed monthly (Spanish Society of Clinical Chemistry). 

Table 1 shows the biochemical and hematological laboratory parameters performed as well as the method and the reference values. Sysmex XN-2000 ^®^ (Roche Diagnostics, Basel, Switzerland) and Advia Centauro XPT^®^ (Siemens, Marburg, Germany) were used to analyze each of those two parameters in turn.

Anemia was defined according to WHO recommendations on the topic, the cut-off for defining anemia and its severity: Hemoglobin level <130 g/L in men and 120 g/L in females [13]. Cut-off values served as criteria for defining hematological and biochemical parameters according to the manufacturer´s recommendations and the reference of the Spanish Society of Laboratory Medicine (SEQCML) [14].

### 2.5. Small-Bowel Biopsy

If serology and the HLA genotype were positive or clinically compatible, patients were scheduled for biopsy. A minimum of 4 small-bowel mucosal biopsies from the distal duodenum and at least 1 from the anatomical duodenal bulb were taken. Histopathologic findings were classified according to the Marsh-Oberhuber classification [15] by an experienced pathologist. 

### 2.6. Diagnostic Criteria 

The diagnosis of CD in adults requires positive serology (particularly -TTG2-IgA) and histological changes on small intestinal biopsy (Marsh type 3a–c) [16]. However, it is expected that the criteria could change in the near future, as studies have suggested that the biopsy-sparing algorithm could apply to adults as well as children [17]. 

### 2.7. Statistical Analysis

Values are expressed in absolute and relative frequencies. To study the relationship between categorical variables, the Chi-square test was used. In the case of relationships between continuous variables correlations were determined, and contrasted if they were significantly different from zero, by using the Correlation Test. A propensity score matching (PSM) procedure was performed to select CD patients adjusted for age and sex with proportion 1:2 against non CD patients. Logistic regression was performed to study the effect of different factors on CD. Results are shown as odds-ratios and their confident intervals. All analyzes were performed with the statistical software R 3.6.3 R Core Team [18]. *p* value of <0.05 was adopted for a significant association of variables. 

### 2.8. Ethical Considerations

This study complies with the Helsinki Declaration on biomedical research involving human subjects as well as with the new and successive updated versions of the applicable legislation, from 2012 to 2020, including successive revision and approval by the territorial committee (Santiago-Lugo) of Galician Network of Research Ethics Committees (approved and registered code number 2017/327; 2019/098 and the subsequent approval to extend the project until 2022). Data were pseudonymized and analyzed in accordance with the current legislation to ensure the protection of personal data (General Data Protection Regulation, Regulation EU 2016/679, and Spanish Organic Law 3/2018).

## 3. Results

### 3.1. CD Case Positive Rate in the Whole Study Sample 

The flow chart in Figure 1 shows the study population, which comprised 9066 individuals referred to our laboratory for first-time measurement of CD antibodies. Of them, 101 individuals (Table 2) were newly diagnosed with CD (1.10%, 95% CI 0.93–1.35). Of those, 84 individuals had a prior hematologic test (Table 3). Among 8975 individuals without CD, 168 age and sex matched were included (1 case: 2 controls).

We found that the newly diagnosed patients with CD had more than 60 years (mean age ± SD 63.04 ±12.1) and a larger percentage were women (68.32%; ratio female/male 2.11/1). 

### 3.2. Serology and HLA Genotype

Mean TTG2-IgA titers were 206 ± 15 U/mL. 90.44% of patients had EMA-IgA titles between 1:10–1:40 and 1:640–1:2560. Serology assessment did not detect patients with IgA deficiency. All positive TTG2-IgA titers were confirmed with high titers of EMA-IgA. 8 individuals (9.52%) presented TTG2-IgA values below the reference limit. 

### 3.3. Intraepithelial Lymphocytosis and/or Villous Atrophy and Crypt Hyperplasia of Small-Bowel Mucosa

Biopsy specimens were taken from 88.12% of patients and analyzed by an expert pathologist. Most patients (76.24%) had marked or complete atrophy (MARSH 3a–c). 

Biopsy was not performed on 11.88% of patients (Figure 1) due to patient refusal, medical recommendation, or having started gluten free diet due to severe symptoms or high positivity in markers. In these cases, the diagnosis of CD was confirmed by presenting typical manifestations of celiac disease, high titters of IgA serum celiac disease antibodies, HLA-DQ2 or DQ8 genotypes and response to the gluten-free diet [16] (Figure 1). 

### 3.4. Clinical Findings at CD Diagnosis

The clinical expressiveness of the disease at the time of diagnosis is shown in Table 2 (gastrointestinal and no hematological extra-intestinal manifestations) and 3 (hematological manifestations). The associated pathologies were mainly autoimmune disorders (17.82%). Chronic abdominal pain, constipation, distended abdomen or irritable colon were the prevalent intestinal CD manifestations (26.73%). 

Since it is of the utmost importance to recognize CD in individuals with non-gastrointestinal symptoms [19,20], we evaluated these manifestations and in particular hematological ones. 

Among the extra-intestinal CD features hematologic ones were more prevalent. Anemia was the most commonly presenting extra-intestinal manifestation (Table 3). A total of 28.57% of newly diagnosed patients with CD were anemic per WHO definition [13]. The distribution of anemia adjusted by age and sex showed that it is more common in patients aged >80, who showed 57.1% and 50% in female and male, respectively (Figure 2). Regarding the anemia severity (WHO criteria) [13], 29.16% of them exhibited moderate (grade II) to severe (grade III) anemia, whereas 70.8% showed mild anemia (grade I) (Table 3).

Anemia in newly diagnosed patients with CD is attributed to multifactorial etiology [19]. In this study, the most prevalent anemia-related deficiencies were ferritin (66.7%) and iron (63%), followed by folic acid (16.6%) and VitB12 (6.1%) (Table 4).

The most remarkable data was that 58.4% of new CD patients exhibited high >14.3% RDW. This cut-off was used according to the reference intervals for our automated hematology analyzer Sysmex XN [21].

### 3.5. Comparative Hematological Features between Seropositive and Seronegative CD Patients 

Figure 1 and Table 3, Table 4 and Table 5 depict differences in hematological and other biochemical biomarkers between newly diagnosed patients with CD and without CD, matched by age and sex.

As Table 3 presents, there were no statistically significant differences between patients who presented CD and those who did not present CD in terms of anemia, low hematocrit, leukopenia and thrombocytopenia. 

On etiological evaluation of anemia, ferritin in patients newly diagnosed with CD and iron in those without CD were the main anemia-related factors as a result of multivariate logistic regression (Table 4).

Since evidence has been provided that RDW could be a marker of CD [8,9], its diagnostic performance has been evaluated, but RDW is an age-dependent parameter, being elevated from the age of 60 [22,23]. In this study high >14.3% RDW was exhibited by 58.4% newly diagnosed adult patients versus 35.2% of those not diagnosed with CD, matched by age and sex, respectively (significate differences at *p* < 0.001). 

As MCV is a parameter used to RDW calculi and also is an age-dependent parameter (elevated from the age of 60) [23], it was analyzed (Table 3).

It was noted that the presence of high RDW (OR 1.151, 95% IC 1.046–1.275) is associated to CD (Figure 3). 

As TTG2-IgA and atrophy grade III (a-c) are the two main CD diagnosis criteria [16] and previous studies found an association between the degree of atrophy and high RDW values [24,25], we have analyzed the association between RDW with atrophy as well as RDW with TTG2-IgA. We have performed an ROC curve to verify the relationship between the degree of atrophy (3a–c) and the RDW, and as can be seen in Figure 4 and Table 6, there was a significate association (*p* = 0.007). By contrast, TTG2-IgA was not associated with RDW (*p* = 0.605). 

## 4. Discussion

In this study, we analyzed the usefulness of a hematologic parameter as a possible marker of CD. RDW, as a part of the automated routine lab test, is employed in daily clinical practice. According to the objectives of this study, we first discussed the main results related to the CD case positive rate in middle and late adulthood and the main clinical manifestations. Thereafter, we analyzed the use of RDW to detect individuals with CD. 

The estimated prevalence of CD in middle and late adulthood is in the range of 0.7% and 2.45% [5,26,27,28,29,30,31], becoming increased four-fold over the last 22 years. According Fueyo-Diaz et al. [30] (northeast of Spain, Aragón) the prevalence of CD in primary care setting range from 0.34% (45–59 years old) to 0.12% (≥90 years old); being 0.27% between 60 and 74 years old. Other study in the Spanish tertiary care setting (northeast of Spain, Catalonia) reported decreasing CD prevalence from middle to late adulthood with a slight increase after age 80 [31]. In our study in a tertiary hospital from the northwest of Spain (Galicia) the case positive rate was 1.10%. Other European countries such as Finland showed higher prevalence of CD in the elderly than has been observed in adults [32] with 2.70% for biopsy-proven CD and seropositivity. That difference may be due to several factors. Firstly, a low index of detection in primary care setting [30]. Secondly, a prolonged seronegativity in elderly with newly detected CD [33]. Thirdly, a heterogeneous clinical pattern [1,3,5] with subtle [1] or few symptoms [3]. Additionally, associated disorders might contribute to occult or underlying CD [34].

Extra-intestinal manifestations configuring the clinical picture of CD in middle-and late adulthood are anemia, micronutrients deficit, neuropathy, etc. Anemia and abnormal laboratory tests are common in late adulthood [5,34]. Anemia was the most prevalent extra-intestinal manifestation in our study (28%). According to the WHO criteria regarding the classification of public health significance of anemia [13], this percentage represents a moderate significance (normal, 4.9% or lower; mild, 5.0–19.9%; moderate, 20.0–39.9%; severe, ≥40%). 

In addition, newly diagnosed CD patients could exhibit various deficiency states, resulting in a large number of manifestations and co-morbidities, including anemia, osteopenia, fractures, neuropathy, etc. The most frequently described deficiencies include iron, folic acid and VitB12, which reflect the loss of absorptive surface area and functional capacity. The estimate deficiencies depend on age, among other factors. Adults can exhibit iron (12–82%), folic acid (20–30%) and Vit B12 (8–75%) deficiency [35]. By comparing the results obtained in our study to those corresponding to a recent study done by Shiha et al. (Tertiary hospital, Royal Hallamshire Hospital, Sheffield United Kingdom) [3], lower percentage was observed for VitB12 (6.1% vs. 10.8–20.2%) and folic acid (16.6% vs. 23.2–28.6%); being higher iron (63% vs. 29.3–37.3%) and ferritin (63.7% vs. 12.1–20.2%) deficiencies in our study. Additionally, we performed a multivariate study of all biochemical parameters that may be related to anemia in CD and non-CD patients. Only ferritin could be considered as the main anemia related factor in CD patients. 

As consequence of isolated or mixed deficiencies, adult and elderly CD patients may suffer extra-intestinal manifestations. These are, among others, neuropathy [36], psychiatric [37] or musculoskeletal disorders [38]. The prevalence of neuropathy ranges from 0% to 39% with an increased prevalence in older and female patients [36]. In the case of psychiatric disorders reaches up to 16.8% in adults [39]. Late-adulthood (CD individuals (*n* = 104, mean age 63.0 years) can exhibit cognitive impairment (mean reaction time task, 621.2 ± 124.0 milliseconds) perceived anxiety (31.0%) or depression (58.4%) [40]. In our study, we observed a prevalence of 7.94% psychiatric disorders (anxiety, depression, and cognitive impairment). With respect to musculoskeletal disorders, reduced bone density [30], increased osteopenia or osteoporosis (from 4.9% at 35–64 years to 23.2% ≥65 years) [3] have been reported. Consequently, increased prevalence of fractures is a common manifestation [38]. In our study, we observed a prevalence of 7.92% in musculoskeletal disorders, including osteopenia and repetition fractures.

Several studies analyzed the potential clinical usefulness of RDW as a risk and prognostic factor in the diagnosis of a number of different diseases (dementia, cerebral infarction, cardiovascular disease, rheumatoid arthritis diseases, chronic obstructive pulmonary disease, coronavirus disease, pneumonia, diabetes, gastrointestinal diseases, etc.) [41,42,43].

With regard to RDW in CD, the studies are scarce [8,9,24,44,45,46,47,48]. To our knowledge, this is the first study that specifically examined RDW in middle and late adulthood newly CD diagnosed patients and age-sex matched controls. PSM was used. Our results could be due to the fact that of having CD, although it could not exclude other causes not embraced by PSM.

Several studies showed RWD increase in adults with CD by using 14% as cut-off value. Sategna et al. [8] in Italy showed an RDW increase in 53.7% of adults with CD diagnose and in 28.6% of adults without CD (individuals with inflammatory bowel disease and diseases other than malabsorption). These authors reported absence of correlation between RDW values and histological scores. Balaban et al. [47] (Rumania) (34 newly diagnosed CD adult patients, 34 age and sex matched controls with irritable bowel syndrome (IBS) and 16 treated CD patients) observed elevated RDW (>14%, normal values under 14%) in 79.41% of the newly CD patients in comparison to IBS and CD-treated patients (17.65% and 37.5%, respectively). Additionally, one study in adults from Denmark (57061 individuals (26 years old, *n* = 706 CD antibody positive) (29 years old, 56.655 CD antibody negative)) showed that RDW was high in 14.4% and 3.7% of CD antibody positive and negative individuals, respectively [44]. Similarly, the study by Harmanci et al. [24] (*n* = 49 newly diagnose CD, mean age 38, Turkey) showed increased RDW in 89% of newly diagnosed CD individuals. Further, Yeşil et al. (61 patients with ulcerative colitis, 56 patients with Crohn’s disease and 44 healthy volunteers, 38–40 years age, Turkey) observed that the percentage of patients with high RDW was significantly higher in CD patients compared to those with ulcerative colitis or controls [48]. 

In our study, we compared newly diagnosed CD patients with non-CD ones, age-sex matched. We used >14.3% as a cut-off value of high RDW in accordance with the reference intervals for our automated hematology analyzer Sysmex XN [21]. High RDW >14.3% was seen in 58.4% of newly diagnosed CD patients comparing to 35.2% of those without CD (significate differences at *p* < 0.001). It must be emphasized that high RDW was associated to CD (OR 1.151, 95% IC 1.046–1.275) in our study.

RDW is considered as a sensitive marker of nutritional deficiency that alters not only red blood cell production but also their maturation. Thus, after 6 [24] or 12 [8,46] months of gluten-free diet, RDW decreases alone [42] or in parallel to restoring normal serum iron, VitB12, folate [8] or ferritin [47] levels, MCV values [47] and EMA seroconversion [8]; reflecting dynamic changes of RDW and its utility as an index of clinical efficacy of treatment. In our future research, we will analyze RDW along with gluten-free-diet in our middle-late adulthood CD patients.

It is well-known that TTG2-IgA and atrophy grade III (a-c) are both required for the diagnose of CD [16]. RDW has been postulated as a predictor of villous atrophy in CD patients (number = 49, median age 38 year old) at a cut-off level of 17.25% [24]. It must be noted that, in this study, we observed a significant association between higher RDW and atrophy grade III, AUC 0.810 (0.702–0.917). Additionally, we found an association between high RDW and CD. Newly CD diagnosed patients had a hematological study available previously for diagnosis, showing high RDW. Thus, we suggest that RDW could be a new aid of CD screening. 

## 5. Conclusions

We suggest that RDW can be used as a predictive factor of CD and potential screening aid of CD.

## Figures and Tables

**Figure 1 ijerph-20-00066-f001:**
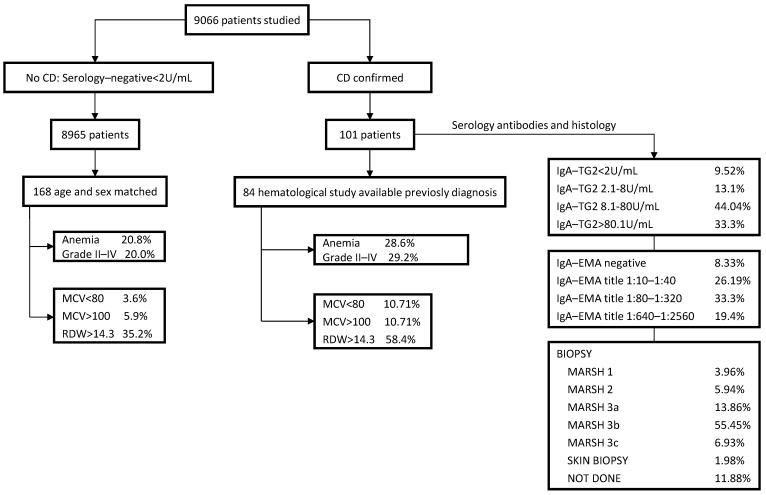
Flow chart of CD diagnosis. Abbreviations: IgA EMA, endomysial IgA antibodies; MCV, Mean cell volume; RDW, Red blood width; TTG2-IgA, Tissue transglutaminase antibody (IgA).

**Figure 2 ijerph-20-00066-f002:**
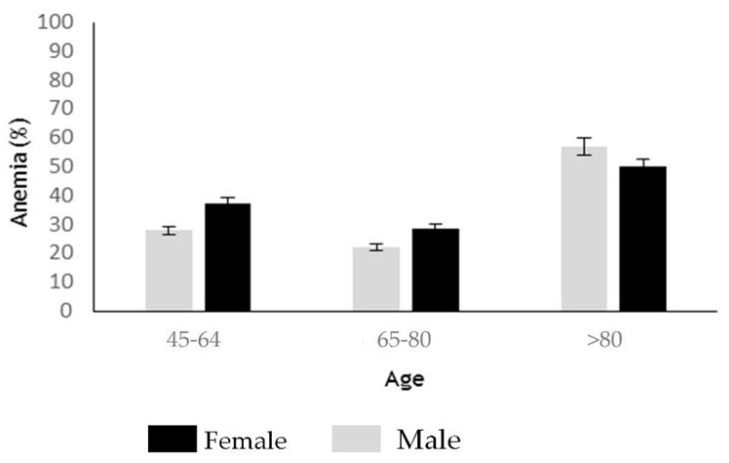
Anemia in patients newly diagnosed with CD, adjusted for age and sex. Data is shown as the mean ± SD. Anemia was established according to the WHO criteria [13]. Abbreviations: CD, Celiac disease; SD, Standard deviation; WHO, World Health Organization.

**Figure 3 ijerph-20-00066-f003:**
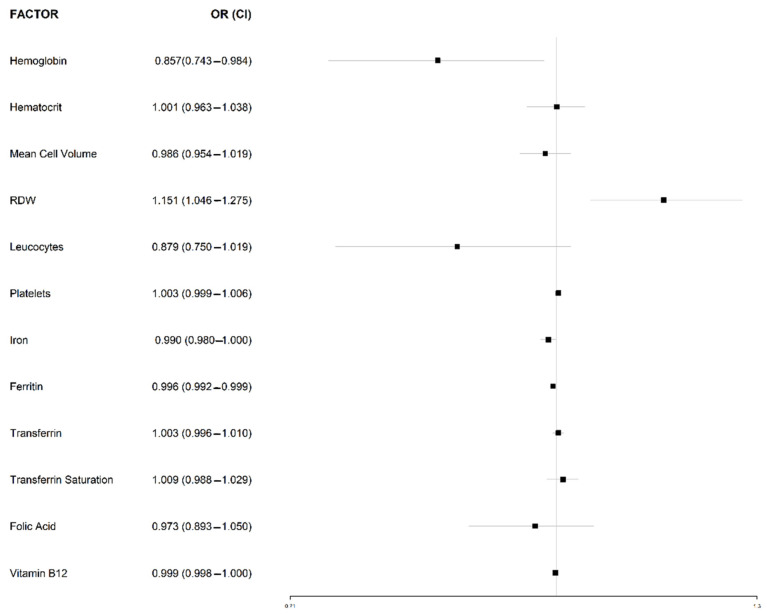
Association between hematological parameters and CD. Logistic regression for comparison of patients newly diagnosed with CD and without CD. Odds ratio with 95% confidence interval. Abbreviations: CI, Coefficient interval; Hct, Hematocrit; MCV, Mean Cell Volume; OR, Odd ratio; PLT, Platelets; RDW, Red cell distribution width; WBC, Leucocytes.

**Figure 4 ijerph-20-00066-f004:**
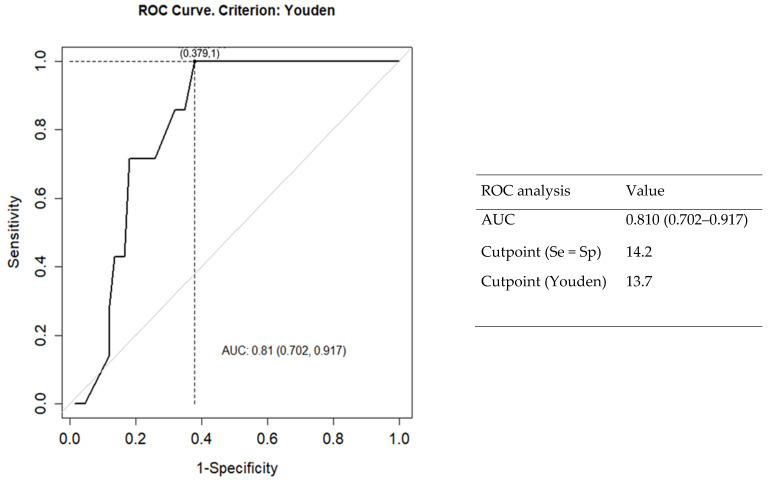
ROC curve to verify the relationship between the degree of atrophy and RDW. Abbreviations RDW, Red cell distribution width: ROC, Receiver operating characteristic.

**Table 1 ijerph-20-00066-t001:** Biochemical and hematological parameters.

	Test	Method	[Rf] Reference Value
Hematological parameters	Hemoglobin concentration	Cyanide-free SLS	>130 g/L men >120 g/L women
	Hct	HDF	0.383–0.486 L/L men 0.355–0.449 L/L women
	MCV	^a^ MCV Cal	81–99 fL
	RDW	^b^ RDW Cal	11.6–14.3%
	WBC	FFC	3.4–9.6 × 10^9^/L
	PLT	HDF	135–317 × 10^9^/L men 157–371 × 10^9^/L women
Biochemical parameters			
	IS	COL	65–175 mg/dL men 50–170 mg/dL women
	IF	IA	22–322 ng/dL men 10–291 ng/dL women
	TfR	IA	250–380 mg/dL men 215–365 mg/dL women
	TS	^c^ Cal	15–50% men 20–50% women
	FA	CL	>5.4 ng/mL grey zone 3.4–5.4 ng/mL, deficiency <3.4 ng/mL
	VB12	AEC	211–911 pg/mL

Abbreviations: AEC, Acridinium ester chemiluminescence; Cal, calculated; C, chemiluminescence; COL, colorimetry; Hct, Hematocrit; HDF, Impedance measurement by means of hydrodynamic focusing; FA, Folic acid; IF, Ferritin; FFC, fluorescence flow cytometry; IA, immunoassay; IS, Iron serum; MCV, Mean Cell Volume; PLT, Platelets; RDW, Red cell distribution width; SLS, sodium lauryl sulphate; TfR, Transferrin; TS, transferrin saturation; VB12, Vitamin B12; WBC, Leucocytes. ^a^ MCV in fL = (Hematocrit %)/(RBC × 10^12^/L) × 10^3^; ^b^ based on both the width of the distribution curve and the mean cell size. 1SD/MCV × 100 where SD and MCV are the standard deviation of the mean cell size and Mean Cell Volume, respectively; ^c^ iron (Transferrin × 1.41) × 100.

**Table 2 ijerph-20-00066-t002:** Clinical and demographic characteristics of patients newly diagnosed with CD.

Demographic characteristics	Mean ± SD	Median (IQR)	n = 101	%
Female			69	68.32
Male			32	31.68
Age	63.04 ±12.1	62 (52.3–73)		
**Gastrointestinal manifestations**			42	50.49%
Chronic diarrhea/malabsorption			10	18.81
Chronic abdominal pain, constipation, distended abdomen or irritable colon			27	26.73
Gastritis, recurrent vomiting, esophagitis and hepatopathy			5	4.95
**Extra-intestinal manifestations no hematological**			50	
Cutaneous/mucosal manifestations: herpetiformis dermatitis, vitiligo			7	6.93
Neurological manifestations: headache and cognitive impairment			6	5.94
Neuropsychiatric manifestations: depression/anxiety			12	11.88
Arthritis/arthralgia and decreased bone mineralization, repetitive fractures			8	7.92
Chronic fatigue, diminished appetite			4	3.96
Others (kidney, cardiac, gynecologic)			6	5.94
**Associated endocrine diseases:** Hypothyroidism, diabetes type II			7	6.93
**Familiar screening**			4	3.96
**Associated autoimmune diseases**			18	17.82
Autoimmune thyroid dysfunction			4	3.96
Systemic autoimmune disease (Sjogren’s syndrome and others)			11	10.89
Organ-specific autoimmune diseases			3	2.97

Abbreviations: CD, Celiac disease; IQR. Interquartile range; SD, Standard deviation.

**Table 3 ijerph-20-00066-t003:** Hematological parameters. Comparison between patients without CD and patients newly diagnosed with CD.

Hematological Parameter		No CD (*n* = 168)	CD (*n* = 84)	*p* Value
Anemia				
	No	133 (79.2)	60 (71.42)	0.171
	Yes	35 (20.8)	24 (28.57)	
^a^ Anemia grade				
	I	28 (80.0)	17 (70.83)	0.576
	II-III	7 (20.0)	7 (29.16)	
Hct				
	Normal value	130 (77.4)	61 (72.61)	0.687
	Low/moderate value	38 (22.6)	23(27.38)	
MCV				
	Macrocytic	6 (3.6)	9 (10.71)	<0.024
	Normocytic	152 (60.5)	66 (78.57)	
	Microcytic	10 (5.9)	9 (10.71)	
RDW				
	Normal value	109 (64.8)	35 (41.6)	<0.001
	High > 14.3%	59 (35.2)	49 (58.4)	
WBC				
	Normal value	165 (98.2)	81(96.42)	0.380
	leukopenia	3 (1.8)	3 (3.57)	
PLT				
	≥150 × 10^9^/L	157 (93.5)	77 (91.66)	0.603
	<150 × 10^9^/L	11 (6.5)	7 (8.3)	

The participants were screened for celiac disease antibodies as shown in Methods. The sample consisted of 9.066 individuals. 84 newly CD diagnosed patients had a hematological study available previous to diagnosis. Among 8975 seronegative, 168 age and sex matched were included. For each hematological biomarker, the comparison between CD and no CD individuals with the proportion below or above the reference interval is shown. Values are expressed in absolute and relative frequencies. Differences are calculated with the Chi-square test. ^a^ WHO grade: I, mid 95–109 g/dL; II, moderate 80–94 g/L; III, severe 65–79 g/L; IV, life-threatening <65 g/L. Abbreviations: CD, Celiac disease; Hct, Hematocrit; MCV, Mean Cell Volume; PLT, Platelets; RDW, Red cell distribution width; WBC, Leucocytes.

**Table 4 ijerph-20-00066-t004:** Anemia-related factors.

**No CD**	**Factors**	**OR (CI)**	***p* Value**
	IF	0.99 (0.98–1.00)	0.556
	tRf	0.99 (0.96–1.01)	0.456
	TS	1.10 (1.00–1.24)	0.061
	FA	1.01 (0.85–1.21)	0.883
	VitB12	0.99 (0.99–1.01)	0.364
	IS	0.93 (0.86–9.81)	0.023
**CD**	**Factors**	**OR (CI)**	***p* Value**
	IF	1.04 (1.01–1.09)	0.025
	tRf	0.96 (0.91–1.01)	0.245
	TS	0.56 (0.26–0.92)	0.058
	FA	1.00 (0.76–1.25)	0.937
	VitB12	1.00 (0.99–1.00)	0.181
	IS	1.27 (1.01–1.78)	0.079

Multivariate logistic regression. Abbreviations: CD, Celiac disease; FA, Folic acid; IF, Ferritin; IS, Iron serum; TfR, Transferrin; TS, transferrin saturation; VitB12, Vitamin B12.

**Table 5 ijerph-20-00066-t005:** Biochemical parameters. Comparison between no CD and newly CD diagnosed individuals.

Biochemical Parameter		No CD ^a^ (n/%)	Newly CD Diagnosed CD ^a^ (n/%)	*p* Value
	Normal	29 (26.4)	20 (37.0)	0.160
IS	Low value	81 (73.6)	34 (63.0)	
	Normal	25 (21.5)	18 (33.3)	0.099
IF	Low value	91 (78.5)	36 (66.7)	
	<250	26 (32.9)	14 (32.5)	0.820
TfR	>380	2 (2.5)	2 (4.6)	
	250–380	51 (64.6)	27 (62.9)	
	<18%	19 (24.0)	7 (16.2)	0.316
TS	≥18%	60 (76.0)	36 (83.8)	
	Normal	56 (78.9)	22 (61.1)	0.025
FA	Grey zone	15 (21.1)	14 (38.9)	
	Deficit	2 (2.8)	6 (16.6)	
	≥200	69 (98.6)	31 (93.9)	0.222
VitB12	<200	1 (1.4)	2 (6.1)	

^a^ Data available that were requested according to clinical criteria to establish the presence or absence of anemia and the anemia type. Abbreviations: FA, Folic acid; IF, Ferritin; IS, Iron serum; TfR, Transferrin; TS, transferrin saturation; VitB12, Vitamin B12.

**Table 6 ijerph-20-00066-t006:** Association between RDW with atrophy grade III.

Atrophy			
	No	Yes	*p* Value
RDW	13.2 (12.8–13.5)	14.7 (13.7–17.0)	0.007 *

Values expressed as medians and interquartile ranges. *p* value calculated with the Mann-Whitney U test. * Significant association. Abbreviations: RDW, Red cell distribution width.

## Data Availability

Data was obtained from Galician Healthcare Service (SERGAS), and are not available.

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
