# Peer review of "Red Cell Distribution Width as a Predictive Factor of Celiac Disease in Middle and Late Adulthood and Its Potential Utility as Celiac Disease Screening Criterion"

_ijerph, 2022, doi:10.3390/ijerph20010066_

Round 1

Reviewer 1 Report

Nicely written paper, I only have minor grammar issues with line 57-61.

Author Response

"Por favor, vea el archivo adjunto"

Reviewer 2 Report

Please find the comments in a separate file below.

Author Response

"Por favor, vea el archivo adjunto" 

Reviewer 3 Report

I reviewed the article IJERPH 2043621. Authors provide a very detailed elaboration of clinical and laboratory characteristics of 101 newly diagnosed CD patients (among 9066 tested - as referred to a tertiary center with a suspicion on CD). My knowledge of CD is really a very general one - that of an average MD. However, I have no doubt that a detailed display of clinical and lab features is of interest for specialists in the field , and possibly broader public. Therefore, I can only comment on some methodological aspects that need to be amended/revised.

1. On several occassions, authors metion RDW in a context of being a "risk factor" (for CD). Such a claim (related to RDW or to any of the other considered clinical or laboratory feature) - is completely misplaced and should be removed: this was a cross-sectional study. The value of RDW as taken at the work-up could have been due to CD and not a cause of CD!. Hence, unless you have a cohort of people without CD (but with known RDW values, which, however, change almost on daily basis) - and record incident CD, where high RDW precedes (and not coincides) with CD diagnosis - you can not and should mention RDW in a context of a "risk factor".

2. In the Discussion, authors conclude that they observed a "prevalence of CD of 1.1%". This is based on the fact that 101/9066 evaluated patients were positively diagnosed with CD. 1.1% here is NOT prevalence. It could be denoted as "case-positive rate", i.e., proportion of those in whom diagnosis was explicitly verified among all those who were referred specifically for this type of evaluation. Prevalence is a population-based term: hence, if you are certain that over this period of 10 years you have not omitted any incident CD patient, then your ten-year (cumulative) prevalence is 101 patients with denominator being the total population of your region times 10 years (to get subjects years of observation), or, simply, you may claim a 10-year prevalence as 101/total number of population (or adult population) in the catchment area. 

3. In this setting, RDW can only be assessed as a potential "diagnostic aid" (or, maybe, screening aid). The way to deal with this quetions is to undertake ROC analysis and search for a cut-off with optimal combination of sensitivity and sensitivity - as you did. But, you could aslo "walk up and down" the values to detect cut-off with highest sensitivity or specificity or whatever you prefer. One thing, clearly, is simply to evaluate sensitivity/specificity of RDW >upper limit of the normal range in your institution. Finally, since this was a cohort of consecutive patients, it is completely justified to use these various sensitivities and specificities and calculate PPV and NPV (for CD and for atrophy or whichever you prefer).

4. The part about "maching" is unclear. You mention PS matching, then age-sex matching...it is confusing. There were 101 CD patients and 9066-101 non-CD patients. I am certain that, e.g., exact matching on age (e.g, 2 or 5-year bins) and sex could yield 101 CD patinets matched to more than 101 "controls". But, by matching on sex and age, and not matching on comorbidities (for example, anemia //and, thus, likely RDW and range of other hematological indicators, could be associated with a range of conditions) or other factors - one does not achieve much. The CD vs. non-CD difference, in this respect, should not be considered as "independent" (of a range of confounding factors that were not accounted for). And if PS was used, how do you draw a PS on 2 covariates?. ..As I said, for sure, even exact matching on a range of covariates (allows 1 case to be matched to 1 or more controls and vice-versa) for sure would have resulted in that all 101 cases could be matched to at least 1 control (each). If PS is what you prefer, then provide (in e.g., supplement) the model for PS, then elaborate the PS-based matching; show distribution of PS across CD and controls etc. (as it is accustomed in any study that uses PS).

5. A huge number of univariate comparisons and P-values were reported..which are mostly not needed. If there is a claer-cut numerical difference, then it is obvious. If it is minor, but p-value is low - you should have in mind the fact that with  some 20 statistical tests (or more) that you did, your actual p-value ("study-wise") was 1-0.9520, that is 0.642.

In brief- show numerical comparisons, skip the series of univariate tests.

Your work will be more straightforward and convincing.

Author Response

"Por favor, vea el archivo adjunto" 

Round 2

Reviewer 2 Report

Review of the manuscript tilted “Red cell distribution width as a risk factor of celiac disease in middle and late adulthood and its potential utility as celiac disease screening criterion” by Cabo del Riego et al.

There are still unresolved issues in the manuscript.

Introduction

Line 49. The Authors did not clarify the matter of anemia diagnostics. I still do not know on what criteria diagnosis of anemia was based – WHO criteria or WHO criteria and ferritin criteria? The combined criteria are for iron-deficiency anemia, not for anemia in general. This is very important to clarify this.

Line 50. The issue has not been clarified.

Material and methods

Line 114. See remark to line 49 in the introduction.

Information on informed consent is still missing in the text. I expect clear indication about this. Did patients give their informed consent to participate in the study or not?

Results

Table 3. I asked the Authors to perform additional analysis and present concrete values for parameters, not categories. I am still interested in median values (and IQRs) for these parameters in both groups.

Table 5. The same remark as in Table 3.

Author Response

Versión PDF

Reviewer 3 Report

Regarding the revised manuscript:

1. English is not my native language and I do not feel really qualified to comment on it...but it seems to me that some minor language polishing would be welcome..(for example, I am not sure that the wording "middle adulthood" really "exists" in the English language in the meaning of..e.g., middle-aged adutls).

2. The authors have replaced the word "risk factor" with "predictive" through-out the text and in the title. Again, with my lack of English knowledge, I am not sure that this word reflects adequately the actual work...RDW here was evalauted as a "diagnostic aid"...i.e., a  simple laboratory finding that might help one diagnose CD. If  the word "predictive" bears this same meaning - then OK; but "predictive" could be also viewed in the sense of "predicting future events", which here would be inappropriate (since the study is cross-sectional).

3. I agree that the authors may, if they wish so, show a large number of p-values from a series of univariate tests. But their claim that "experimentwise" type 1 error rate is not affected by this fact because "outcomes were independent" - simply is not true. For example, all hematological lab findings may resonably be considered "mutualy related". I do not suggest that the authors should employ any type-1 error rate multiplicity adjustment - I only state that such tests are not very meaningful and the authors should simply claim that there were "numerical differences", i.e., that some values were higher in one and lower in the other subset of patients (numerically) - but should not refer to the shown P-values as "evidence" of "significance", "meaningfulness" or "importance".

4. I still claim that a PS based on two variables (age and sex) with nearest 1:2 matching does not remove confounding bias. I agree that the authors are free to display differences between age and sex-matched (based on a two-variable PS) patient subsets, but it is the interpretation of these differences that matters - it could be due to the fact of having or not having CD, but it could also be due to many other reasons (causes) not embraced by the PS. This is something that should be taken into account when discussing the study findings.

Author Response

 versión PDF
